# Nanosensors Based on Bimetallic Plasmonic Layer and Black Phosphorus: Application to Urine Glucose Detection

**DOI:** 10.3390/s24155058

**Published:** 2024-08-05

**Authors:** Fatima Houari, Mohamed El Barghouti, Abdellah Mir, Abdellatif Akjouj

**Affiliations:** 1Laboratory of Advanced Materials Studies and Applications (LEM2A), Physics Department, Faculty of Science, Moulay Ismail University of Meknes, B.P. 11201 Zitoune Meknès, Morocco; f.houari@edu.umi.ac.ma (F.H.); or mohamed.elbarghouti@gmail.com (M.E.B.); a.mir@umi.ac.ma (A.M.); 2Faculty of Medicine and Pharmacy of Beni Mellal, Sultane Moulay Slimane University, M’ghila Campus, 23030 Beni Mellal, Morocco; 3Univ. Lille, Institute of Electronics, Microelectronics and Nanotechnology, UMR CNRS 8520, FST, Department of Physics, 59655 Villeneuve d’Ascq, France

**Keywords:** SPR biosensor, bimetallic film, glucose level detection, sensitivity, biomedical sensing, black phosphorus layer

## Abstract

This paper presents a new biosensor design based on the Kretschmann configuration, for the detection of analytes at different refractive indices. Our studied design consists of a TiO_2_/SiO_2_ bi-layer sandwiched between a BK7 prism and a bimetallic layer of Ag/Au plasmonic materials, covered by a layer of black phosphorus placed below the analyte-containing detection medium. The different layers of our structure and analyte detection were optimized using the angular interrogation method. High performance was achieved, with a sensitivity of 240 deg/RIU and a quality factor of 34.7 RIU^−1^. This biosensor can detect analytes with a wide refractive index range between 1.330 and 1.347, such as glucose detection in urine samples using a refractive index variation of $10^{-3}$. This capability offers a wide range of applications for biomedical and biochemical detection and selectivity.

## 1. Introduction

Recently, the surface plasmon resonance (SPR) technique has become a versatile and widely used nanotechnology for studying molecular interactions in real time and without labeling [1,2,3]. This technique has emerged as the most promising detection and biosensing method due to several advantages, including the very good sensitivity and quality factor (QF), ease of manufacture and use, lower manufacturing costs, and simple, rapid identification compared with conventional sensors [4,5,6,7]. These top-tier features make SPR-based biosensors highly attractive and widely used for many biosensing applications, such as molecular biology, biochemistry and pharmaceuticals [8,9], human blood groups [10], glucose concentration [11,12], bacteria [13], gas detection [14], food and the environment [15], and temperature measurement [16], among others.

Due to our remarkable ability to detect variations in the refractive index (RI), we based our SPR design on Kretschmann configurations [17]. Various approaches have been developed to improve the performance of the SPR biosensor studied, generating considerable interest in the design of SPR biosensors with optimal characteristics [18,19,20]. By considering various parameters such as the materials introduced and their thicknesses in the envisaged SPR nanostructure, surface plasmons are reinforced by bimetallic film and nanomaterials to further trap light [21].

Our objective is to investigate the design of highly sensitive SPR biosensors capable of detecting biological, biochemical, and biomedical analytes with very small variations in the refractive index of the sensing medium. To excite the surface plasmon modes in the proposed design, we used a coupling prism, which operates on the attenuated total reflection (ATR) principle [22]. Much research work has focused on improving the performance of SPR biosensors, in particular on improving sensitivity and the QF. Recently, M. Wang et al. [23], J. Banerjee et al. [24], and N. Mudgal et al. [25] have presented SPR sensor models incorporating Ag and Au bimetallic films and 2D materials such as BaTiO_3_, WS_2_, and graphene. These models gave sensitivity and QF values of 192.5 deg/RIU–26.37 RIU^−1^, 185 deg/RIU–24.44 RIU^−1^, and 116.67 deg/RIU–37.87 RIU^−1^, respectively. In addition, B. Dey et al. [26] proposed an SPR sensor based on Au/Al film and WS_2_/graphene layers and, it showed a sensitivity of 208 deg/RIU with a QF of 223.66 RIU^−1^. B. Karki et al. [11] proposed an SPR sensor based on two Ag films intercalated by an MXene layer (Ag/MXene/Ag) and covered by a ZnO/graphene hybrid layer, achieving a sensitivity of 184 deg/RIU with a QF of 27.23 RIU^−1^.

In this paper, we propose an original hybrid multi-layer structure BK7/TiO_2_/SiO_2_/Ag/ Au/BP/SM (see Figure 1), which was computationally developed for biomedical biosensing. We used a bimetallic Ag/Au film to take advantage of the fine plasmon resonance properties of silver and the stable optical and chemical characteristics of gold [27]. Additionally, we integrated a TiO_2_/SiO_2_ hybrid bi-layer between the prism and the metal due to their remarkable effect in efficiently light-screening the SPR design [28]. Finally, a layer of the promising BP 2D nanomaterial was deposited on our nanostructure to enhance sensitivity and bind biomolecules by improving analyte adsorption. This nanomaterial, which has attracted researchers due to its enormous electrical properties and small optical band gap, has been used in a wide range of biosensing applications [4]. Recently, S. C. Kishore et al. [19], B. Karki et al. [21], and L. Wu et al. [29] have reported some BP monolayers in the SPR structure. On this basis, they showed, respectively, that the use of the BP layer enhances the SPR sensor’s analyte detection range, improves detection performance, and presents a biochemical recognition element. The results are discussed mainly in terms of sensitivity and the QF, with the main motivation being to model and optimize a highly sensitive SPR biosensor for biomedical applications. The optimized performance parameters of different nanostructures to obtain the optimal design are as follows: S = 144 deg/RIU and QF = 37 RIU^−1^ for BK7/Ag/Au/SM (1), S =176 deg/RIU and QF = 35.3 RIU^−1^ for BK7/TiO_2_/SiO_2_/Ag/Au/SM (2), and S = 240 deg/RIU and QF = 34.7 RIU^−1^ for BK7/TiO_2_/SiO_2_/Ag/Au/BP/SM. This represents an improvement of 66.67% and 36.36% over the conventional SPR sensors (1) and (2), respectively.

A high concentration of glucose in the human body can lead to a number of illnesses, mainly diabetes and health problems related to the heart, eyes, kidneys, and other organs. Diabetes is a chronic disease that occurs when the pancreas does not produce enough insulin or when the body cannot use the insulin it does produce efficiently. According to World Health Organization (WHO) statistics, diabetes affects around 422 million people worldwide and kills more than 1.6 million annually [30,31]. The refractive index of the urine sample from a person with diabetes differs from that of an average healthy person due to changes in glucose concentration (g/dL). The SPR biosensor we propose is highly sensitive and capable of detecting small changes in urine glucose concentration, corresponding to an RI variation of 0.001. Better sensitivity (243.7 deg/RIU) and a good QF (31.32 RIU^−1^) were obtained in comparison with other SPR biosensors.

The manufacturing cost of our SPR sensor design is also normal and acceptable for two reasons: (i) The majority of the materials making up the design are inexpensive and required in small quantities (such as BK7, TiO_2_, SiO_2_, and Ag). Other materials have a medium cost, but are used in acceptable quantities (5 nm Au and four monolayer BP) [32,33]. (ii) The simplicity of our proposed SPR sensor design allows for easy fabrication using several specific and readily available techniques. For instance, thin-film deposition methods such as physical vapor deposition, atomic layer deposition, or epitaxy can positively influence the cost [34].

Thanks to their efficiency and experimental feasibility using simple and common manufacturing nanotechnologies, the proposed SPR biosensor offers considerable potential for the detection of biochemical and biomedical analytes, as well as in the field of biomedical analysis.

## 2. Modeling, Refractive Index, and Performance Parameter

### 2.1. Optical Properties of the Materials’ Nanolayers

The proposed Kretschmann configuration for the SPR biosensor is analyzed using a seven-layer nanostructured design, whose nanostructure is illustrated in Figure 1. In this proposed plasmonic design, we have taken a BK7 prism, on which TiO_2_/SiO_2_ nanocomposite layers are deposited. Our SPR biosensor is based on the Ag/Au/BP/SM hybrid nanomaterial multilayer. In this proposed SPR nanostructure, the bimetallic Ag/Au layer is the plasmonic material for surface plasmon (SP) excitation, and covered by a few BP layers. The BP layer represents the binding and recognition layer of the analyte in the SM.

A homogeneous light source with a wavelength of 633 nm and transverse magnetic polarization (TM) is used to excite the SPs in the proposed SPR design. This light source is launched from one side of the BK7 prism, and the reflected light is detected from the other side. When a light beam is totally reflected at the boundary, it generates an evanescent wave, which interacts with the plasma wave provided the metal has an appropriate thickness of a few tens of nanometers.

When the SPR occurs, the excitation wave vector ($k_{light}$) and the wave vector in the Ag metal film ($k_{SP}$) are equal [35,36] ($k_{light}=k_{SP}$), and one obtains the SPR angle as:(1)$$\theta_{SPR}=\arcsin\sqrt{\frac{n_{m}^{2}.n_{s}^{2}}{n_{p}^{2}\left(n_{m}^{2}+n_{s}^{2}\right)}},$$
where $n_{p}$, $n_{m}$, and $n_{s}$ represent the RIs of the prism, metal layer and SM, respectively. Depending on various environmental conditions such as atmospheric pressure, vibration, temperature, humidity, and radiation, which can influence the behavior of a material, it has been found that the optical properties of the materials making up the SPR nanostructure are affected considerably as a function of diverse effects such as the wavelength [37].

#### 2.1.1. BK7-Prism Refractive Index

In the current SPR biosensor, we designed several configurations to optimize the effect of various layers on detection efficiency. A prism with a low refractive index offers high sensitivity [8,38], so we considered a BK7 prism. The refractive index of the BK7 as a function of wavelength was calculated using the following equation [19,39]:(2)$$n_{BK7}={\left(\frac{1.03961212\lambda^{2}}{\lambda^{2}-0.00600069867}+\frac{0.231792344\lambda^{2}}{\lambda^{2}-0.0200179144}+\frac{1.01046945\lambda^{2}}{\lambda^{2}-103.560653}+1\right)}^{1/2},$$
where λ is the wavelength of incident light.

#### 2.1.2. Optical Properties of TiO_2_ and SiO_2_ Films

The SPR nanostructures for the proposed nanoplasmonic sensor were based on a prism and coupled to a TiO_2_/SiO_2_ nanocomposite layer, as shown in Figure 1.

The TiO_2_ and SiO_2_ layers were selected for their excellent optical properties in the visible and near-infrared range, designed to enhance nanoplasmonic performance. The refractive indices of the bilayer are $n_{TiO_{2}}$ = 2.5837 for titanium dioxide and $n_{SiO_{2}}$ = 1.4570 for silicon dioxide (at 633 nm) [40,41].

#### 2.1.3. Bi-Metallic Films

In order to describe the optical properties of the bi-metallic film, the optimized Ag film thickness was taken to be 45 nm, covered by a 5 nm thin Au film. According to the Drude–Lorentz model, the wavelength dependence of the refractive index for the metals in question can be expressed as [21,42]:(3)$$n_{m}^{2}\left(Ag,Au\right)=\varepsilon_{m}=1-\frac{\lambda^{2}\lambda_{c}}{\lambda_{p}^{2}\left(\lambda_{c}+i\lambda \right)},$$
where $\lambda_{p}$ and $\lambda_{c}$ represent the plasma and collision wavelength. The numerical values of $\lambda_{p}$ and $\lambda_{c}$ are 1.4541 × 10^−7^ m and 1.7614 × 10^−5^ m for silver and 1.6826 × 10^−7^ m and 8.9342 × 10^−6^ m for gold, respectively.

At a wavelength of 633 nm, the refractive indices of the Ag and Au bimetallic layer for the biosensor computation have been calculated from Equation (Equation 5).

#### 2.1.4. Black Phosphorus (BP) Nanomaterial Layer

The sixth layer is BP, a two-dimensional nanomaterial, which can be deposited as an affinity layer to bind and enhance the density of detected analytes, thus improving SPR biosensor performance (see Figure 1). Thanks to BP’s excellent characteristics, it has a positive effect on the uptake capacity of biomolecules and is used to protect the SPR nanostructure against oxidation, thereby increasing its stability [21,43]. The thickness of the BP monolayer is $d_{BP}$ = 0.54 nm, and its refractive index at a wavelength of 633 nm is $n_{BP}$ = 3.5 + i0.01 [43].

#### 2.1.5. Sensing Medium Layer

In new SPR biosensor models, the detection of extremely small variations in the refractive index is seen as the primary objective. The present nanodesign aims to detect different biomedical refractive indices. The RI of the SM lies between 1.330 and 1.335, corresponding to different sample solutions that can be prepared by mixing different solute mass ratios in phosphate-buffered saline (PBS) solution [27]. Different biological urine solutions with low glucose concentrations generally have RIs between 1.335 and 1.347. The refractive index of this layer is computed by n_*s*_ = $1.33+\Delta n_{s}$, where $\Delta n_{s}$ represents the variation in the refractive index due to the biomedical molecular change analyzed [4,8,11,18].

### 2.2. Theory Modeling and Performance Parameters

The plasmonic spectral response has been evaluated by numerical calculations of the optical properties of SPR nanoconceptions using the finite-element method (FEM) [44,45,46]. This method solves Maxwell’s equations by discretizing space into small finite elements, enabling the electromagnetic field at each point in space–time to be evaluated according to the material parameters and initial boundary conditions [47,48,49].

The proposed SPR nanostructures are excited by an incident electromagnetic light wave of wavelength λ = 633 nm generated in the BK7 prism. It is cast in the Oxy plane at a variable incidence angle (θ) with respect to the Oy axis. This excitation wave is TM polarized, and its associated electric field vibrates along the x and y axes.

In attenuated total reflection (ATR) configurations, the intensity reflectance coefficient (R(ω)) spectrum can be calculated using the S-parameters acquired from the numerical simulation presented in the following equation:(4)$$R\left(\omega \right)=|S_{11}{|}^{2}$$

In this study, the evaluation of the detection performance of an SPR biosensor includes the treatment of its sensitivity and enhancement, its detection accuracy, and its quality factor; the results achieved are higher and suggest superior proposed biosensor capabilities.

Adsorption of biomolecules in the sensing medium of the SPR design modifies its RI. The most significant performance parameter of SPR biosensors is sensitivity, calculated by the shifting of the plasmon resonance angle $\Delta \theta_{SPR}$ in degrees over the variation of the RI of the sensing medium $\Delta n_{s}$ in Refractive Index Units (RIU). The other parameter is the sensitivity improvement S_%_, obtained from S and S_0_, the optimized sensitivity and the conventional sensor, respectively [3,38,50], which are given by:(5)$$S=\frac{\Delta \theta_{SPR}}{\Delta n_{s}}\left(deg/RIU\right)$$
(6)$$S_{\%}=\frac{S-S_{0}}{S_{0}}\left(\%\right)$$

The detection accuracy (DA) is obtained from the inverse of the maximum width at half-maximum (FWHM) [50]:(7)$$DA=\frac{1}{FWHM}\left(deg^{-1}\right)$$

Finally, the quality factor (QF) is calculated using the following equation [3,50]:(8)$$QF=\frac{S}{FWHM}=S\times DA\left(RIU^{-1}\right)$$

## 3. Results and Discussions

### 3.1. Optimization of the Proposed Multilayer SPR Design

Recently, silver (Ag), one of the most frequently used plasmonic metals in SPR nanostructures, has been shown to produce finer SPR peaks than other metals [37]. It, therefore, appears to be a preferable choice for SPR biosensors used in biological, biochemical, and medical detection due to its very good quality factor. However, since the Ag metal film oxidizes in the presence of chemical and biological molecules [51,52], engineering an SPR biosensor nanostructure with an Ag/Au bimetallic layer is a preferable approach for two reasons: (i) Ag is subject to oxidation with the environment to make it chemically stable, and the nanostructure can be coated with an Au film overlying the Ag film as a protective layer, as illustrated in Figure 1; (ii) it improves the detection performance of the SPR biosensor.

In this section of our study, we present the results of the optimization carried out to determine the optimum thickness of the Ag metal film with a 5 nm Au film. With a thickness $d_{Ag}$ = 45 nm, as shown in Figure 2a, the minimum SPR reflectance intensity curve is almost zero, as shown in Figure 2b. Figure 2c shows the evolution of the SPR resonance angles, which shift towards larger angles as $d_{Ag}$ increases from 20 to 60 nm with 5 nm Au, for the two RI of the detection medium 1.330 and 1.335. Figure 2d illustrates the effect of varying the Ag thickness on the detection sensitivity of the proposed design. An increase in the sensitivity of the bimetallic (Ag/Au) biosensor is obtained, with the best value being S=144 deg/RIU when the Ag layer thickness is 45 nm, representing a 500% improvement over the sensor based on mono-metal Ag (without the Au film). At this thickness, the QF is also significant, reaching 37 RIU^−1^.

We used the TiO_2_/SiO_2_ hybrid layer in our SPR biosensor nanostructure, BK7/TiO_2_/ SiO_2_/Ag/Au because TiO_2_/SiO_2_ has a greater plasmonic effect, enabling it to trap light efficiently. This efficient light trapping generates more surface plasmons, resulting in an increase in the SPR angle. This increase in the SPR resonance angle leads to enhanced SPR biosensor performance [40,53]. To optimize the TiO_2_/SiO_2_ layer, we set the Ag film at 45 nm and the Au film at 5 nm.

First, we optimized the thickness of the TiO_2_ layer. Figure 3a shows the reflectance spectra as a function of the incidence angle for two RIs of the sensing medium at $d_{TiO_{2}}$ = 30 nm. We found that the sensor sensitivity increases from 144 deg/RIU without TiO_2_ to 168 deg/RIU with the TiO_2_ layer and a QF = 36.7 RIU^−1^. In the second step, we added the SiO_2_ layer. As shown in Figure 3b, the reflectance intensity curves for $d_{TiO_{2}}$ = 30 nm/$d_{SiO_{2}}$ = 20 nm indicate that the sensitivity also increases with the deposition of the optimized SiO_2_ layer, reaching a value of 176 deg/RIU and QF = 35.3 RIU^−1^. Therefore, for the optimum TiO_2_/SiO_2_ thickness, we chose values where the minimum reflectance is relatively lower and the sensitivity higher, which is a more important condition for obtaining a typical nanostructure to apply and study the biosensor application.

Figure 4a illustrates the distribution of electric and magnetic field intensity as a function of the prism–interface distance for the proposed nanostructure, optimized when the RI of the sensing medium is 1.330. It is well known that, when the reflectivity intensity takes its minimum value, the biosensor is in plasmon resonance (here, $\theta_{SPR}$ = 72.32 deg). At this point, the electric and magnetic field intensity approaches its maximum value, as shown in Figure 4b. Figure 4a also shows that the electric field strength increases in the TiO_2_/SiO_2_ layer due to the nanoplasmonic effect occurring near this layer, thereby increasing the sensitivity of the proposed design. Additionally, the bimetallic layer significantly enhances the field strength and exhibits a peak at the Au/sensitive medium interface for analyte detection [3,8].

In this section, we have noted that the inclusion of a BP layer between the bimetallic nanofilm and the sensing medium offers an improvement in performance. Moreover, it is known that the exceptional optical properties of two-dimensional BP nanomaterial have made it widely used in SPR sensing designs over the past few years [54,55]. BP exhibits high mobility, a direct band gap, a very high surface-to-volume ratio, and greater adsorption of biomolecules [56,57].

First, in optimizing the number of BP monolayers in our proposed SPR design, we kept the optimized nanostructure without BP as follows: TiO_2_ (30 nm)/SiO_2_ (20 nm)/Ag (45 nm)-Au (5 nm)/BP (varies). Figure 5a,b show the reflectance spectra as a function of the angle of incidence for different BP monolayers (*L*) (L=0 to 7). The reflectance intensities are calculated for two RIs of the detection medium: 1.330 [Figure 5a] and 1.335 [Figure 5b]. There is a large shift in the plasmonic resonance angle for both media as *L* increases from 0 to 7.

Figure 5c shows the optimized parameters for sensitivity, the DA, and the QF, which are, respectively, 240 deg/RIU, 0.132 deg^−1^, and 34.7 RIU^−1^, corresponding to L=4, with an improvement in sensitivity of 36.36%, when the BP layer is inserted compared with the optimized design without BP, as shown in Figure 5d.

Figure 6 shows the distribution of the electric field intensity as a function of the normal prism–interface distance. The field intensity first increases with the thickness of the Ag metal film and in the Au film, reaching its maximum value of 1.58 × 10^5^ V/m at the BP/sensitive medium interface. This observation confirms the generation of SPs in the SPR structure [18]. Subsequently, this maximum field intensity decreases exponentially in the sensing medium.

In addition, we calculated the penetration depth (PD) of the biosensor concerned with the size of the biomolecules detected. The PD is the distance traveled by the field wave of maximum value decreasing up to 37% (i.e., 1/e, where *e* = 2.718) [4]. The PD measured in this 156.5 nm design is better than that already obtained [3,4,29,58].

The SPP modes in the studied BK7/TiO_2_/SiO_2_/Ag-Au/BP/SM design are crucial for the fabrication of a highly sensitive and specific biosensor, as shown in Figure 6b–d. Figure 6b shows 2D and 3D tracings of the electric field intensity distribution of the 1D curve presented in Figure 6a. Figure 6c,d show the 2D and 3D tracings of the *x*- and *y*-components of the electric field in the SPR nanostructure. We observed strong intensity localization at the design interface. To clarify the intensity advantage at the design interface, it is important to maximize the interaction between surface plasmons and molecules to enhance detection sensitivity.

### 3.2. Biodetection of Glucose Concentration in Urine Samples from Patients with Diabetes

The biosensing of glucose in urine samples using SPR biosensors is indeed a very important application in nanobiotechnology and biomedicine, particularly for monitoring patients suffering from diabetes [12]. To detect the glucose concentration in urine samples, several parameters of the SPR biosensor need to be optimized for the best selectivity and sensitivity. To achieve these two important parameters, the proposed SPR sensor must maintain the specificity of the biological receptor; often, an enzyme such as glucose oxidase is crucial for specific binding to glucose, minimizing interactions with other substances in the urine [59]. Experimental conditions, such as temperature, pH, and ionic strength, must be strictly controlled to promote this specific binding [60]. Careful preparation of the sensor surface is essential to reduce non-specific interactions, and selectivity must be validated by testing complex samples and comparing them with pure glucose solutions [61,62]. These include the choice of a well-optimized SPR design, the nanomaterial of the detection surface, adjusting the excitation wavelength, and the immobilization chemistry to bind the biomolecules to be identified [63,64]. Let us assume that a patient with diabetes presents a variation in his/her urinary glucose level. Similarly, we need to apply urine samples to the biosensor detection medium in order to detect the presence and concentration of glucose in these samples. This glucose level fluctuation is due to a change in the value of the refractive index of the detection layer.

Our SPR design detects a broad RI range of the glucose mixture in urine, evolving from n_*s*_ = 1.335 to 1.347, corresponding to a concentration variation of 0–0.015 g/dL to 10 g/dL. Figure 7a shows the reflectance spectra of different concentrations analyzed in this work. Figure 7b shows the variation of the SPR resonance angle as a function of the RI, corresponding to the different concentrations; the solid black squares represent the calculated data, and the red line represents the linear curve fit, which is applied to the calculated data. It can be seen that, as the sample concentration increases from 0–0.015 g/dL to 10 g/dL, the RI also changes in line with the respective change in glucose concentration. Figure 7b illustrates how the RI changes as a function of this variation in glucose concentration, which can be used to identify glucose levels in a urine sample. It can be seen that the SPR angle ($\theta_{SPR}$) varies linearly from 78.48 deg to 81.36 deg. After applying linear curve fitting to the data obtained, we found the following linear fitting equation:(9)$$\theta_{SPR}\left(n_{s}\right)=243.7\times n_{s}+78.48$$$\theta_{SPR}$ = 78.48 deg represents the plasmon resonance nail, at the origin of the RI of the detection medium (here, the reference RI) n_*s*_ = 1.335.

Here, the slope of the line of fit given by Equation (Equation 8) is 243.7 deg/RIU, which represents the average detection sensitivity of the concentration level in urine samples from persons with diabetes.

Table 1 shows the variation in refractive index as a function of changes in glucose concentration, and the performance parameters obtained, for determining glucose levels in urine samples. For an RI variation of analyte Δn_*s*_ = 0.001, the measured sensitivity, DA, and QF are 240 deg/RIU, 0.128 deg^−1^, and 30.85 RIU^−1^, respectively. This gives a 90.48% improvement over the reference SPR biosensor [11].

In this section, we demonstrate that our SPR design could be of significant interest for the detection of biochemical and biomedical analytes with an RI between 1.330 and 1.347. In particular, our SPR biosensor design provides better performance in terms of sensitivity and the quality factor compared to published SPR sensors, as presented in Table 2. Additionally, our proposed design offers simplicity and the ease of experimental verification. Consequently, our SPR design remains the most sensitive biosensor for biomedical diagnostics and biosensing of glucose in human urine samples.

### 3.3. Experimental Feasibility of the Proposed SPR Biosensor Chip

Our study is based on the numerical modeling of the multilayer SPR biosensor based on a bimetallic film. Figure 1 shows the SPR design of the proposed device. At this stage, we discuss the realization of the proposed SPR biosensor for a practical implementation. The recommended SPR biosensor requires a BK7 prism as the initial substrate. First, the BK7 prism substrate must be thoroughly cleaned to remove any contaminating particles that might affect design performance. Next, a thin TiO_2_/SiO_2_ nanocomposite bilayer is applied to the cleaned flat prism surface using a technique based on sol–gel fabrication methods [32]. The thickness of the desired and precisely optimized TiO_2_/SiO_2_ bilayer is $d_{TiO_{2}}$ = 30 nm/$d_{SiO_{2}}$ = 20 nm to meticulously control the deposition parameters. Next, a bimetallic film of Ag/Au is deposited, respectively, on the TiO_2_/SiO_2_ layer using the physical vapor deposition (PVD) technique [65]. To achieve the desired, optimized thickness of $d_{Ag}$ = 45 nm/$d_{Au}$ = 5 nm, we need to fine-tune the deposition parameters and closely monitor the deposition rate. Then, to complete the design of our biosensor, a layer of 2D BP nanomaterials is deposited layer by layer (four monolayers) on the metal film, using the chemical vapor deposition (CVD) technique [66]. Considerable research effort has been devoted to the production of highly pure and stable BP monolayers, requiring the use of diverse techniques such as microscopy, solution stripping, supercritical carbon dioxide assistance, chemical vapor deposition, and plasma etching [67]. Finally, the detection medium is attached to the BP free surface, i.e., the region where biomolecules interact with the device.

For efficient detection of different glucose concentrations in urine samples, with an RI of 1.335 (0–0.015 g/dL) to 1.347 (10 g/dL), a portion of filtered urine is taken as the base sample (natural glucose concentration). Pure glucose is dissolved in distilled water to prepare a concentrated stock solution. Samples are stored at low temperature (4 °C) and analyzed rapidly to avoid the degradation of biochemical components. The additive method is used to add appropriate volumes of this stock solution to urine samples, in order to achieve the desired glucose concentrations (see Table 1).

## 4. Conclusions and Perspectives

In summary, we have modeled and optimized an SPR biosensor based on a nanostructure composed of bimetallic film and nanomaterial layers, BK7/TiO_2_/SiO_2_/Ag/Au/BP/SM, with enhanced performance properties, using the finite-element method (FEM). The effect of adding bimetallic film and a BP layer to our SPR design is to benefit from evolving functionalities such as the biodetection of a wide range of analytes. The design optimization results show that the sensitivity can reach up to 240 deg/RIU and QF of 34.7 RIU^−1^, which is superior to the different sensors based on bimetallic film. In addition, the analysis of the electric field intensity distribution in the proposed biosensor provides valuable information on the sensor’s penetration depth, which is closely related to sensitivity and determines the depth at which target analytes can be detected. The proposed SPR biosensor is capable of detecting different levels of glucose concentrations present in a urine sample with good sensitivity. This is an important and essential factor for the biodetection of low concentrations of different biomarkers. Consequently, the proposed biosensor is well suited to the detection of analytes from diseases associated with kidney and liver dysfunction and for medical diagnostic nanotechnology applications in general.

## Figures and Tables

**Figure 1 sensors-24-05058-f001:**
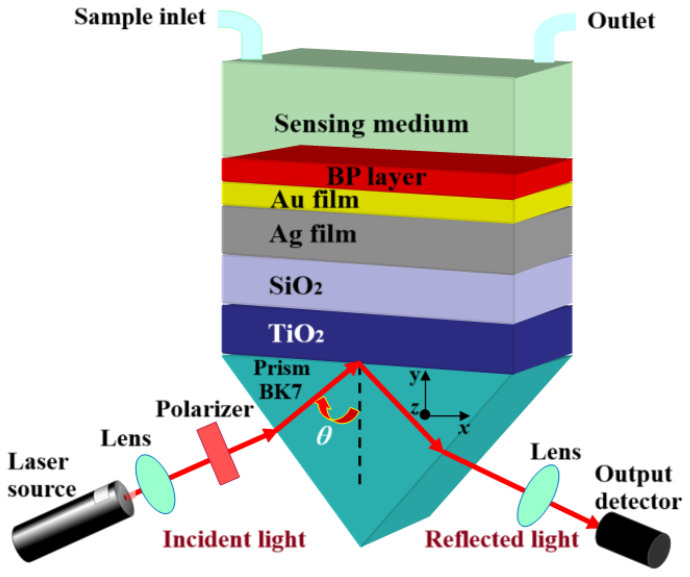
Schematic design of the proposed bimetallic SPR biosensor.

**Figure 2 sensors-24-05058-f002:**
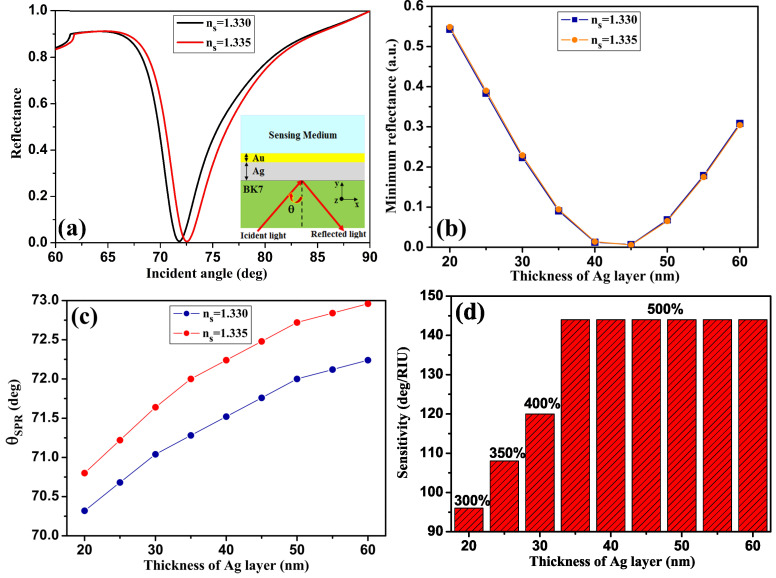
(**a**) Variation of the reflectance with respect to the incidence angle for the refractive index n_*s*_ = 1.330 (black line) and n_*s*_ = 1.335 (red line), (**b**) reflected light intensity versus thicknesses of Ag ($d_{Ag}$), (**c**) evolution of the resonance angle, and (**d**) variation of sensitivity with thickness of Ag layer. For SPR nanostructure with 5 nm of Au.

**Figure 3 sensors-24-05058-f003:**
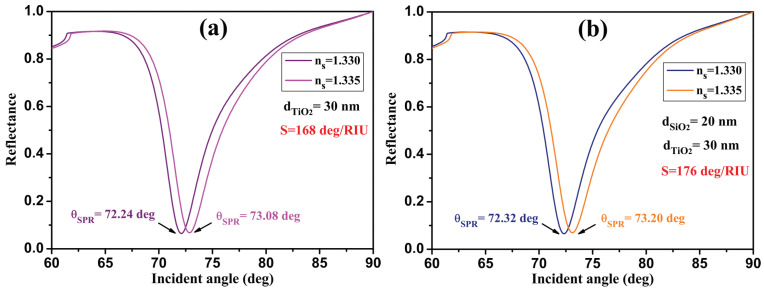
Reflectance intensity as a function of the angle of incidence of the proposed SPR biosensor, BK7/Ag-Au/SM: (**a**) with the TiO_2_ layer and (**b**) with the TiO_2_/SiO_2_ layer, for two sensing mediums n_*s*_ = 1.330 and 1.335.

**Figure 4 sensors-24-05058-f004:**
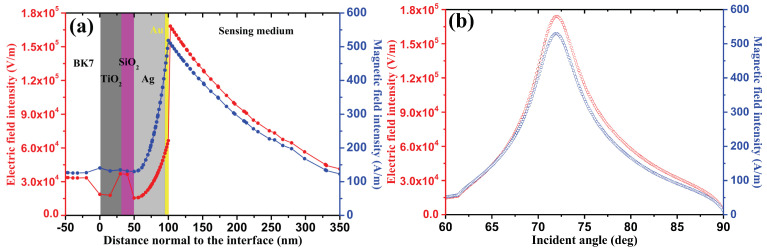
Variation of electric and magnetic field intensity of proposed SPR sensor BK7/TiO_2_/ SiO_2_/Ag/Au, (**a**) in accordance with the distance normal to the interface and (**b**) as a function of the incidence angle at a wavelength of 633 nm, for the RI of detection medium 1.330.

**Figure 5 sensors-24-05058-f005:**
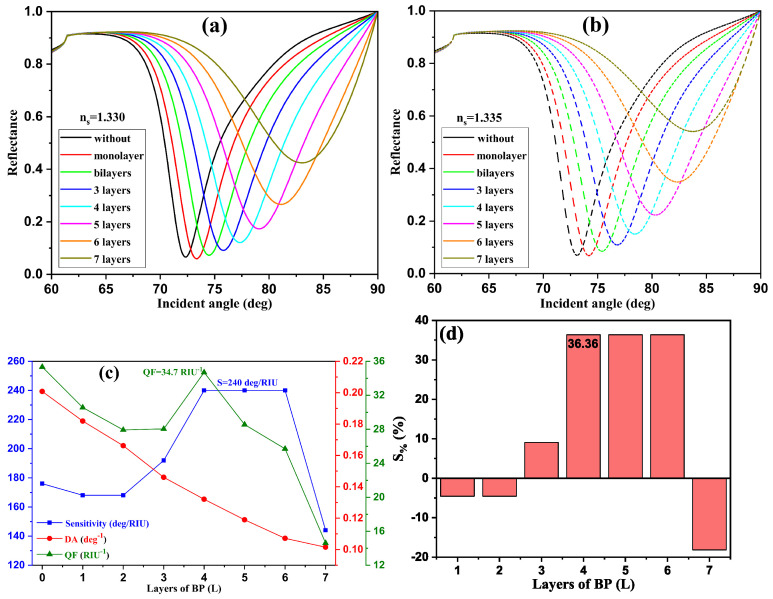
(**a**,**b**) Reflectance intensity curves for different BP monolayers [(**a**) for n_*s*_ = 1.330, (**b**) for n_*s*_ = 1.335]; (**c**) evolution of the sensitivity, DA, and QF; (**d**) sensitivity enhancement as a function of the BP monolayer number for proposed SPR biosensors: BK7/TiO_2_/SiO_2_/Ag-Au/BP/SM.

**Figure 6 sensors-24-05058-f006:**
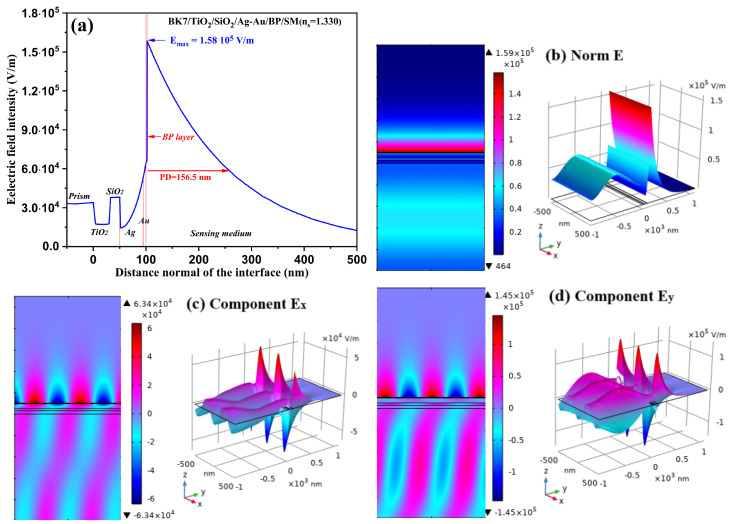
(**a**) The 1D curve of electric field intensity as a function of the distance prism–sensing interface; (**b**) 2D and 3D plots of the distribution of the electric field norm across the optimized SPR design; (**c**,**d**) 2D and 3D plots of SPP modes in the x- and y-component, respectively, of the electric field.

**Figure 7 sensors-24-05058-f007:**
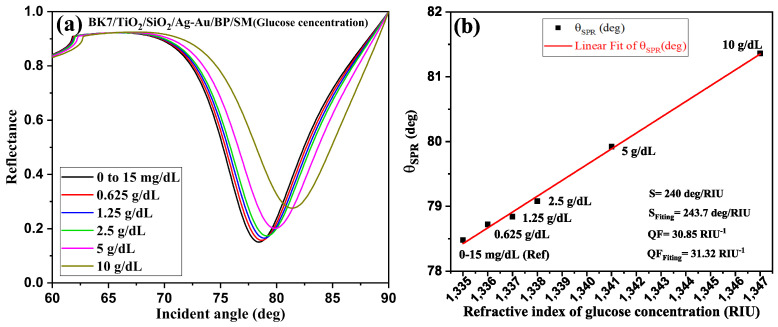
(**a**) Glucose level biodetection in the urine samples using concentration variation; (**b**) resonance angle shift of SPR curves with varying RI of the different glucose concentration level.

**Table 1 sensors-24-05058-t001:** Performance evaluation of biodetection of glucose levels in a urine sample.

Glucose Concentration (g/dL)	Refractive Index	$\theta_{\mathrm{SPR}}$ (deg)	R_*min*_	Sensitivity (deg/RIU)	FWHM (deg)	DA (deg^−1^)	QF (RIU^−1^)
0–0.015	1.335	78.48	0.1503	Ref	7.68	0.130	-
0.625	1.336	78.72	0.1574	240	7.78	0.128	30.85
1.25	1.337	79.84	0.1650	180	7.88	0.127	22.86
2.5	1.338	79.08	0.1731	200	7.98	0.125	25
5	1.341	79.92	0.2009	240	8.19	0.122	29.28
10	1.347	81.36	0.2755	240	8.61	0.116	27.84

**Table 2 sensors-24-05058-t002:** Comparison of the biosensor proposed and previously reported sensors.

SPR Sensor Design	RI Range	Sensitivity (deg/RIU)	DA (deg^−1^)	QF (RIU^−1^)	References
SF11/Ag/Au/WS_2_/graphene/SM	1.332–1.372	192.5	0.137	26.37	Wang et al. Ref. [23]
BK7/Ag/Au/WS_2_/graphene/SM	1.332–1.335	185	0.1321	24.44	Banerjee et al. Ref. [24]
CF2/WS_2_/Ag/Al/WS_2_/graphene/SM	1.330–1.360	208	1.12	223.66	Dey et al. Ref. [26]
SF10/ZnO/Ag/Au/BaTiO_3_/graphene/SM	1.330–1.450	116.67	-	37.87	Mudgal et al. Ref. [25]
BK7/Ag/MXene/Ag/ZnO/graphene/SM	1.335–1.336	184	0.148	27.23	Karki et al. Ref. [11]
BK7/TiO_2_/SiO_2_/Ag/Au/BP/SM	1.330–1.335	240	0.132	34.7	This work
BK7/TiO_2_/SiO_2_/Ag/Au/BP/GC	1.335–1.347	240	0.128	30.85	This work
BK7/TiO_2_/SiO_2_/Ag/Au/BP/GC	1.335–1.341	243.7	0.128	31.32	This work (linear fitting)

## Data Availability

Data are contained within the article.

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
