# Peer review of "Nanosensors Based on Bimetallic Plasmonic Layer and Black Phosphorus: Application to Urine Glucose Detection"

_sensors, 2024, doi:10.3390/s24155058_

Round 1

Reviewer 1 Report

Comments and Suggestions for Authors

This paper presents a SPR sensor design based on the Kretchmann configuration. The studied sensor can detect analytes. It is interesting. However, the structure of the studied sensor is too complex. I do not think this is a promising method for glucose determination. In addition, there are some questions need to be improved.

1. There are many layers, and the sixth layer is BP for the SPR sensor. How about the cost and reproducibility of the prepared sensor? 

2. When abbreviating any terms, spell them out the first time, and use the abbreviation after that. For example, what is QF?

3. What did the authors mean by “increases from 0-0.015 to 10 g/dL” (Line 259)? Please give standard unit for the concentration.

4. The authors claimed that the SPR biosensor need to be optimized for best selectivity and sensitivity. How about the selectivity of the prepared SPR sensor for glucose?

5. How did the authors pave the way for the biodetection of diseases associated with kidney and liver dysfunction?

Comments on the Quality of English Language

There are some mistakes in grammar.

Author Response

Comments 1. There are many layers, and the sixth layer is BP for the SPR sensor. How about the cost and reproducibility of the prepared sensor?

Response 1: We thank the referee for this comment; it has been taken into account in the manuscript.

Firstly, it can be justified by saying that our original hybrid multilayer design BK7/TiO2/SiO2/Ag/Au/BP/SM (see Figure 1), which is being developed for biomedical biosensing. We use an Ag/Au bimetallic film to take advantage of the very fine plasmon resonance properties of silver and the very stable optical and chemical characteristics of gold, and have integrated a TiO2/SiO2 hybrid bilayer between the prism and the metal due to their remarkable light-shielding effect of the SPR design. Finally, a promising BP 2D-nanomaterial layer was deposited on our nanostructure, to enhance sensitivity and bind biomolecules by improving analyte adsorption. This nanomaterial, which has attracted researchers due to its enormous electrical properties and low optical band gap, has been used in a wide range of biosensing applications. Moreover, it is more advantageous for biomedical and biochemical applications.

-The manufacturing cost of our SPR sensor design is also normal and acceptable for two reasons:

(i) the majority of the materials making up the design are less expensive and used in small quantities (such as BK7, TiO2, SiO2 and Ag). Other materials have a medium cost, but are used in acceptable quantities (5 nm Au and 4 monolayer BP) [32,33].  (ii) the simplicity of our proposed SPR sensor design, it is easily fabricated by several specific and available techniques. For example, thin-film deposition methods such as physical vaporization, atomic layer growth or epitaxy can positively influence cost [34]. To date, there is a great deal of international research into the development of 2D nanomaterials (graphene, graphene oxide, TMDCs, MX2 and BP ...) and their production in large quantities and at lower cost.

-For good reproducibility, we rely on the quality of the materials used. Ensuring the homogeneous quality of the black phosphorus and other materials used in the proposed sensor is crucial to reproducibility. Well-controlled and standardized manufacturing processes are essential to obtain reproducible sensors. Characterization and quality control ensure that each sensor produced meets the required specifications. Regular testing and systematic calibration are necessary to maintain reproducibility.

[32] I. Pavlichenko, A. T. Exner, P. Lugli, G Scarpa and P. V. Lotsch, Tunable thermoresponsive TiO2/SiO2 Bragg stacks based on sol-gel fabrication methods.” Journal of intelligent material systems and structures 24 (2013) 2204-2214.

[33] Q. M. Al-Bataineh, A. D. Telfah, C. J. Tavares and  R. Hergenröder,  Modeling and analysis of discrete particle detection in wide-field surface plasmon resonance microscopy. Sensors and Actuators A: Physical 370 (2024) 115266.

[34]  Y. V. Stebunov, O. A. Aftenieva, A. V. Arsenin and A. V. Volkov, Highly sensitive and selective sensor chips with graphene-oxide linking layer. ACS Applied Materials & Interfaces,7 (2015) 21727-21734.

We have added in paragraph and some references numbered 32, 33 and 34 in the Introduction (see page 2, lines 73-80).

Comments 2. When abbreviating any terms, spell them out the first time, and use the abbreviation after that. For example, what is QF?

Response 2: This comment has been taken into consideration in the manuscript.

-Quality Factor (QF)

-Refractive Index (RI)

(see in the abstract in page 1 and in the introduction in page 2).

Comments 3. What did the authors mean by “increases from 0-0.015 to 10 g/dL” (Line 259)? Please give standard unit for the concentration.

Response 3: We thank the referee for this comment; it has been taken into account in the manuscript.

-Assuming the patient suffers from diabetes, the level of glucose concentration in his urine fluctuates. To detect the presence and concentration level of glucose in urine samples, it is necessary to place these samples on the surface of the detection medium. This variation in glucose level is due to the change in the value of the refractive index of the detection medium. As shown in Figure 7(a), the plasmonic response for different glucose concentration values tested 0-0.015g/dL, 0.625 g/dL, 1.25 g/dL, 2.5 g/dL, 5 g/dL and 10 g/dL corresponds to refractive indices of 1.335, 1.336, 1.337, 1.338, 1.341 and 1.348 respectively (see Table 1).

It can be seen that as the sample concentration increases from 0-0.015 g/dL to 10 g/dL, the refractive index (RI) also changes in line with the respective change in glucose concentration. Figure 7(b) illustrates how RI changes as a function of this variation in glucose concentration, which can be used to identify glucose levels in a urine sample. It can be seen that the SPR angle ( ) varies linearly from 78.48 deg to 81.36 deg. (see page 9, lines 277-281).

- The standard unit for glucose concentration in urine can vary, but is generally expressed in one of the following units:

  • Grams per deciliter (g/dL) or Milligrams per deciliter (mg/dL), this unit is commonly used in the USA.
  • Milimoles per liter (mmol/L), this unit is often used in Europe and Canada.

Comments 4. The authors claimed that the SPR biosensor need to be optimized for best selectivity and sensitivity. How about the selectivity of the prepared SPR sensor for glucose?

Response 4: Regarding the selectivity of the proposed SPR sensor for glucose concentration detection, several points need to be considered:

-SPR sensor selectivity is highly dependent on the capacity of the biological receptor to bind the analyte, usually an enzyme such as glucose oxidase, to bind specifically to glucose. Receptors must be selected and possibly modified to minimize interactions with other imperial substances present in the urine [59].

-Experimental conditions, such as temperature, pH and ionic strength, must be rigorously controlled to promote specific binding of glucose to the receptor without reaction from other analytes [60,61].

- The SPR sensor interface must be carefully prepared to reduce non-specific interactions. This can include blocking non-specific analytes, and the use of layers of 2D sensing materials has become a preferred, protective and binding material that prevents non-specific adsorption [62].

-Furthermore, the selectivity of the proposed SPR sensor for glucose concentration detection needs to be optimized for our purpose of use; such as the excitation wavelength and the geometric parameters of the nanodesign, must be optimized to maximize the signal difference between the presence and absence of glucose. In addition, samples containing complex mixtures of analytes must be validated and compared with the responses of pure glucose solutions.

We have added a paragraph and some references numbered 59, 60, 61 and 62 in page 9, lines 257-266.

[59] D. Li, J. Su, J. Yang, S. Yu, J. Zhang, K. Xu and Yu, Optical surface plasmon resonance sensor modified by mutant glucose/galactose-binding protein for affinity detection of glucose molecules. Biomedical Optics Express 8(11) (2017) 5206-5217.

[60] D. A. Gough, J. Y. Lucisano and P. H. Tse, Two-dimensional enzyme electrode sensor for glucose. Anal. Chem. 57(12) (1985) 2351–2357.

[61] M. D. Raicopol, C. Andronescu, R. Atasiei, A. Hanganu, E. Vasile, A. M. Brezoiu and L. Pilan, Organic layers via aryl diazonium electrochemistry: towards modifying platinum electrodes for interference free glucose biosensors. Electrochim Acta 206 (2016) 226–237.

[62] J. Wang, Y. Xu, Y. Song and  Q. Wang, Surface Plasmon Resonance Sensor Based on Fe2O3/Au for Alcohol Concentration Detection. Sensors 24(14) (2024) 4477.

Comments 5. How did the authors pave the way for the biodetection of diseases associated with kidney and liver dysfunction?

Response 5: We thank the referee for this comment.

Our proposed and optimized SPR biosensor is capable of detecting different levels of glucose concentration present in a urine sample with good performance. This study paves the way for the biodetection of diabetes in its early stages.

Our proposal enabled us to detect smaller changes in the refractive index in the sensing medium (0.001), equivalent to a 0.61g/dL variation in glucose concentration, with a thousand-fold improvement in performance (90.48% for sensitivity) over several sensors in the literature (see Table 2 of comparison). By optimizing these technical aspects and validating their results. This is an important and essential factor for bio-detecting low concentrations of different biomarkers.

Consequently, the proposed bio-dispositive is well suited to the detection of analytes from diseases associated with kidney and liver dysfunction, and for medical diagnostic nanotechnology applications in general.

We have added a paragraph in the Conclusion and perspectives.

Reviewer 2 Report

Comments and Suggestions for Authors

This manuscript is devoted to the creation of a new SPE biosensor based on the Kretchmann configuration, for the detection of glucose. The subject of the study is important and relevant. The manuscript is well written, the graphics are carefully designed, but the description of experimental details is omitted. The manuscript can be accepted after responding to the following comments:

1) Examples of the use of black phosphorus layers in sensors should be described in more detail in the introduction.

2) The methods for deposition of SiO2, TiO2, Ag and Au layers of various thicknesses should be described in the experimental part  so that any researcher can repeat them.

3) The method of deposition of layers of black phosphorus should also be described in detail. How was the deposition of various number of BP layers controlled? It is advisable to present some methods for studying BP layers (for example, microscopy).

4) Which solutions were used as sensing media to obtain the data presented in figures? This information should be added to the figure captions and/or to the text. Preparation of all tested solutions should be described in the Experimental section.

5) How were urine samples prepared? Did you use the method of the additives to vary the concentration of glucose in urine? Please describe the technique of urine sample preparation in detail.

Round 2

Reviewer 1 Report

Comments and Suggestions for Authors

The revision is not so bad.

Reviewer 2 Report

Comments and Suggestions for Authors

Authors answered all reviewer's comments. The manuscript can be accepted.